# The development of an operational system for estimating irrigation water use reveals socio-political dynamics in Ukraine

Jacopo Dari[1,2,*], Paolo Filippucci[2], Luca Brocca[2]

[1] Department of Civil and Environmental Engineering, University of Perugia, Perugia, Italy
[2] National Research Council, Research Institute for Geo-Hydrological Protection, Perugia, Italy

*Correspondence to*: Jacopo Dari (jacopo.dari@unipg.it)

**Abstract.** Irrigation is the main driver for crop production in many agricultural regions across the world. The estimation of irrigation water has the potential to enhance our comprehension of the Earth system, thus providing crucial data for food production.

In this study, we have created a unique operational system for estimating irrigation water using data from satellite soil moisture, reanalysis precipitation and potential evaporation. As a proof of concept, we implemented the method at high-resolution (1 km) during the period of 2015-2023 over the area south of the Kakhovka dam in Ukraine, collapsed on June 6, 2023. The selected study area enabled us to showcase that our operational system is able to track the effect of the pandemic and conflict on the irrigation water supply. A significant decrease of 63% and 44% in irrigation water compared to the mean irrigation water between 2015-2023 has been identified as being linked to the collapse of the dam and, potentially, to the COVID-19 pandemic, respectively.

## 1 Introduction

In recent years, Europe has experienced a number of catastrophic events, including the COVID-19 pandemic and the conflict in Ukraine. The effects of these events are far-reaching and impact all sectors of society, including agriculture and food production (Van Tricht et al., 2023). Ukraine is among the largest wheat producers in Europe, and is indeed known as the breadbasket of Europe. It is crucial to comprehend the impact of these catastrophic events on crop production in Ukraine, particularly in the conflict-affected Kherson region. Crop production in the Kherson area south of the Kakhovka dam heavily depends on irrigation that is facilitated by the Kakhovka reservoir. As the Kakhovka dam's collapse on June 6, 2023 is anticipated to have had a significant impact on crop production, it could be evaluated by examining the variability over time of irrigation water use in the area.

Thanks to the advance in satellite technology, as for instance the launch of the Sentinel constellation under the Copernicus Programme, remote sensing has recently enabled the acquisition of irrigation water use measurements (Massari et al., 2021; McDermid et al., 2023), enabling large areas to be monitored in a consistent and equitable manner. This circumstance opens unprecedented perspectives in water resources management over human-altered basins. In fact, irrigation represents the largest component of the anthropogenic water use (Foley et al., 2011; Dorigo et al., 2021), with impacts on several components of the Earth system and social dynamics (McDermid et al., 2023). In general, satellite observations of hydrological variables that can be a proxy of irrigation occurrence are used to estimate irrigation volumes, as long as the condition of a matching between the spatio-temporal resolution of the observational data and the spatial and temporal scales of irrigation dynamics is satisfied (Dari et al., 2022a; Zappa et al., 2022). Specifically, approaches based on satellite soil moisture (e.g., Lawston et al., 2017; Brocca et al., 2018; Dari et al., 2020) and evaporation (e.g., Bretreger et al., 2022; Brombacher et al., 2022; Kragh et al., 2023) products have been developed in recent years. An example integrating both soil moisture and evaporation products is the soil moisture based (SM-based) inversion approach developed by Dari et al. (2023) as an evolution of the SM2RAIN (Soil Moisture to RAINfall) algorithm originally designed to estimate rainfall from satellite soil moisture (Brocca et al., 2014). Preliminary promising results were shown by Brocca et al. (2018) and Filippucci et al.

(2020) by means of coarse resolution satellite and in-situ soil moisture, respectively. Concurrently, a few studies
demonstrated the importance of considering the evapotranspiration term within the algorithm structure together with SM
(Jalilvand et al., 2019; Dari et al., 2020; 2022b). The first implementation with high-resolution satellite soil moisture as an
input has been proposed by Dari et al. (2020). The authors produced a data set of irrigation estimates at 1 km spatial
resolution over a heavily irrigated portion of the Ebro basin, in Spain, covering the period 2011-2017. Recently, the SM-
based inversion approach has been implemented under the European Space Agency (ESA) Irrigation+ project for producing
the first regional-scale, high-resolution data sets over three major basins worldwide (Dari et al., 2023). In a nutshell, the SM-
based inversion approach proved itself to be a useful tool for estimating irrigation water use across scales; the following
natural step is the exploration of the possibility of building an operational system based on it, as currently no operational
services for monitoring large-scale irrigation are available.

In this study, we have developed for the first time an operational system for monitoring irrigation water use with 10 days
latency relying on the SM-based inversion approach forced with operational satellite-based surface soil moisture data and
precipitation and evaporation data from reanalysis.

**2 Study area**

As a proof of concept, the operational system for monitoring irrigation water use from satellite data has been implemented
over a cold semi-arid area (Beck et al., 2018) enclosing a heavily irrigated portion fed by the Kakhovka reservoir on the
Dnipro river (approximate length of 2200 km and average flow at the outlet of 53 km$^3$/year under natural conditions) in
Ukraine, collapsed on June 6, 2023. Under operating conditions, the store capacity was of 18.2 km$^3$, corresponding to an
extent of water surface equal to 2155 km$^2$. We have selected a box of almost 4000 km$^2$ whose extension ranges from
longitude 33.30° to 34.45° and from latitude 46.15° to 46.50°. This is the area fed by the Kakhovsky canal, which originates
just upstream the dam and delivers water to five irrigation districts through an efficient and automated network; the districts
are equipped with a dense system of centre pivot that was mainly realized between the late 1970s and 1980s as part of the
development of the Kakhova system, completed in 1990 (Kuns, 2018) and representing one of the largest irrigated areas in
Europe. The dense system of center pivot irrigation equipment can be observed by visual inspection of satellite maps (see,
e.g., Fig. 1a). For the selected area, the latest version of Global Map of Irrigated Areas (GMIA) (Mehta et al., 2022) reports
peaks up to 60% in terms of percentage of area equipped for irrigation. The data set refers to cells characterised by a spatial
resolution of 5 arc-minutes (about 10 km at the Equator). Reznik et al. (2016) report a percentage of irrigated areas equal to
83.3% of the total available area in 2015. Based on statistical surveys, the main cropping season for cereal and other annual
crops in Ukraine is from May to August (Portmann et al., 2008).

**3 Materials and Methods**

**3.1 The SM-based inversion approach**

Irrigation water use has been estimated through the SM-based inversion approach (Brocca et a., 2018; Dari et al., 2020;
2023) over a time span ranging from January 1, 2015 to September 30, 2023. The core idea behind the method is the
inversion of the soil water balance for backwards estimating the total water input, generally represented by rainfall plus
irrigation. By expressing the soil water balance as:

$$Z^* \frac{dS(t)}{dt} = i(t) + r(t) - g(t) - sr(t) - e(t) \hspace{2cm} (1)$$

where $Z^*$ [mm] is the water capacity of the soil layer, $S(t)$ [-] is the relative soil moisture (i.e., ranging between 0 and 1), $t$
[days] indicates the time, $i(t)$ is the irrigation rate [mm/day], $r(t)$ [mm/day] is the rainfall rate, $g(t)$ [mm/day] is the

drainage term, $sr(t)$ [mm/day] is the surface runoff, and $e(t)$ [mm/day] is the evapotranspiration rate. Eq. (1) is equivalent to:

$$Win(t) = Z^* \frac{dS(t)}{dt} + g(t) + sr(t) + e(t) \qquad (2)$$

where $Win(t)$ is the total amount of water entering into the soil, i.e., rainfall plus irrigation. As thoroughly explained in previous studies by the authors, the following assumptions can be adopted: (i) $g(t) = aS(t)^b$ (Brocca et al., 2014), (ii) $sr(t) = 0$ (Brocca et al., 2015), (iii) $e(t) = F \cdot S(t) \cdot PET(t)$ (Dari et al., 2023). Hence, Eq. (2) can be rewritten as:

$$Win\,(t) = Z^* \frac{dS(t)}{dt} + aS(t)^b + F \cdot S(t) \cdot PET(t) \qquad (3)$$

After estimating the total amount of water entering the soil, irrigation rates can be derived by removing rainfall rates from the total, $i(t) = Win\,(t) - r(t)$. Negative irrigation rates, if any, are imposed equal to zero (Jalilvand et al., 2019). A threshold value for the ratio between weekly estimated irrigation and weekly rainfall equal to 0.2 is adopted to discard negligible irrigation amounts due to random errors.

The parameters $a$, $b$, $Z^*$, and $F$ of Eq. (3) are the model parameters. $a$, $b$, and $Z^*$ have been calibrated against rainfall (i.e., by optimizing the method performances in properly reproducing rainfall amounts) by masking out days with no rainfall rate during the irrigation seasons (hence, potential irrigation days). The F parameter has been set equal to 0.3 as explained in Section 3.2. For further details on the method, the reader is referred to Dari et al. (2023).

## 3.2 Data and processing

The algorithm requires soil moisture, rainfall and potential evapotranspiration (PET) data as an input. We have used Sentinel-1 surface soil moisture observations from the Copernicus Global Land Service (https://land.copernicus.eu/global/products/ssm) (Bauer-Marschallinger et al., 2019) having a spatial resolution of 1 km and 2 to 6 days revisit time depending on the region of interest and the number of satellites available in orbit (2 satellites from October 2016 to December 2021, and 1 satellite from October 2014 to October 2016 and from January 2022 to September 2023 due to failure of Sentinel-1B). Before running the algorithm, the noise in the soil moisture signal has been reduced by computing the Soil Water Index (SWI) according to the exponential filter proposed by Albergel et al. (2008). Precipitation and PET have been obtained from the 5th land reanalysis of the European Centre for Medium-Range Weather Forecasts (ERA5 Land, https://cds.climate.copernicus.eu/cdsapp#!/dataset/reanalysis-era5-land?tab=overview), characterised by a native spatial resolution of 9 km and hourly temporal resolution (Muñoz-Sabater et al., 2021). As PET from ERA5 Land represents pan evaporation, i.e., open water evaporation, we have rescaled the data, i.e., to obtain the same mean value in the common period, by using potential evaporation from the Global Land Evaporation Amsterdam Model (GLEAM v3.6a, https://www.gleam.eu/) (Miralles et al., 2011; Martens et al., 2017) that was used in previous studies (e.g., Dari et al., 2023) but it is not available with short latency. The applied scaling value is equal to 30%. Precipitation and PET data have been resampled over the same spatial grid of Sentinel-1 with the nearest neighbouring approach and aggregated at daily time scale. Therefore, irrigation water use estimates have been produced on the 1 km grid of Sentinel-1 surface soil moisture data with a temporal resolution of 15 days.

## 4 Results and discussion

To assess irrigation water use from satellite observations of soil moisture (or evaporation), the observations must detect the increase in soil water associated with irrigation application. As highlighted in previous studies (see, e.g., Dari et al., 2022a; Zappa et al., 2022), three prerequisites are needed: 1) spatial resolution comparable to irrigated fields, 2) temporal resolution with at least one observation per week, and 3) good accuracy. If any of these conditions are not met, irrigation cannot be

estimated accurately. In this research, Sentinel-1 soil moisture data was used as characterised by an actual spatial resolution of 1 km and a revisiting time of 4 to 5 days in the study area, in accordance with the specifications.

To assess the reliability of the Sentinel-1 soil moisture product for estimating irrigation, we have computed the mean soil
moisture over the months from June to August, corresponding to the main watering season, during the whole study period. Figure 1 displays a satellite map of the study area (on the left) and the mean soil moisture (on the right). The correlation between the regions with center pivot irrigation equipment and those with mean relative soil moisture exceeding 45% is clearly apparent. Consequently, we can confidently stipulate that Sentinel-1 soil moisture data is capable of detecting the irrigation signal in space with good precision.


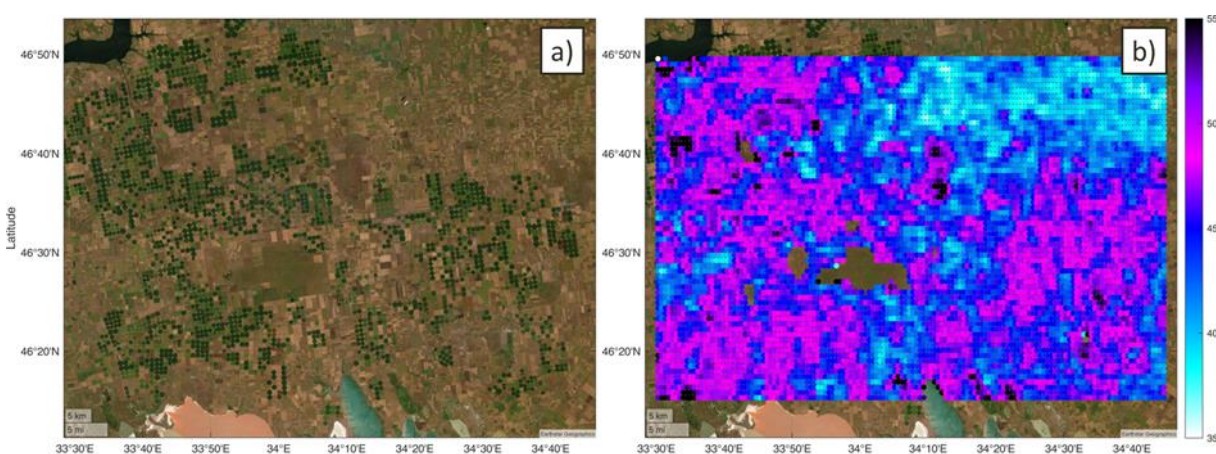

**Figure 1: a) Satellite map of the study area (Earthstar Geographics), green circles represent the center pivot irrigation equipment. b) Mean soil moisture map as obtained from Sentinel-1 CGLS product. Areas with mean values greater than 45% are in a very good agreement with the presence of center pivot irrigation equipment.**


The temporal accuracy assessment is then carried out from 2015 to 2023 by comparing the spatial average of soil moisture and precipitation over pixels in which the mean soil moisture is greater than 45% (Fig. 1). To facilitate the visual assessment, we have shown here a single year, all years are available in the Appendix (Fig. A1). Figure 2 shows the analysis for the year 2017, with also true-colour Sentinel-2 images acquired on May 17, and June 29, 2017. Starting from mid-May to
the end of June, the increase of soil moisture in Fig. 2a is not associated with precipitation events (see Fig. 2b) and it should be attributed to irrigation water application as can be also inferred from Sentinel-2 images showing that the center-pivot areas are much greener at the end of June with respect to mid-May. Conversely, the non-irrigated green areas in May are brown-coloured at the end of June due to the absence of precipitation. The Appendix shows the climatology of soil moisture and precipitation, including modelled soil moisture data also (Fig. A2). The strong increase in soil moisture for the month of
June and the slow decrease in July and August is another point of evidence for irrigation application. The simulated soil moisture does not show this signal as an irrigation module is not included.

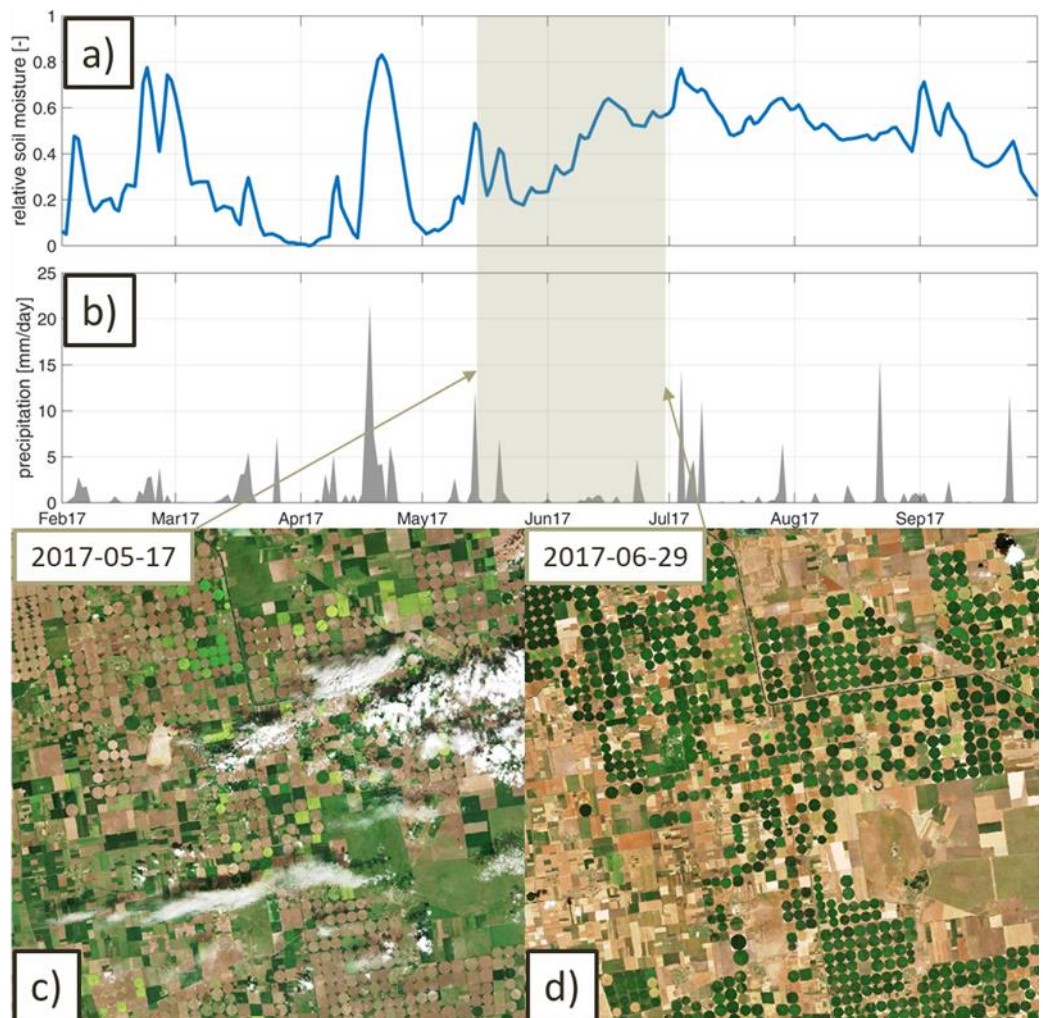

**Figure 2: The time series in the top panels represent the spatial mean relative soil moisture from Sentinel-1 CGLS product (a) and precipitation from ERA5 Land (b) in the period February to September 2017. The spatial average is computed for pixels with average soil moisture greater than 45% as shown in Fig. 1. The two maps in the bottom panels show true-colour Sentinel-2 images acquired on May 17, and June 29, 2017 (obtained from Sentinel Hub EO Browser service). Starting from mid-May to the end of June, the increase of soil moisture in the panel (a) is not associated with precipitation events (b). Therefore, it is likely due to irrigation application as can be also inferred from Sentinel-2 images showing that the center-pivot areas are much greener at the end of June with respect to mid-May.**

As Sentinel-1 soil moisture has been considered suitable to estimate irrigation water use in the study area, the approach developed by Dari et al. (2023) has been applied for all irrigated pixels (for a total of 4059 pixels) in the investigated area. For each pixel, the irrigation water use each15 days has been computed and monthly aggregated values are shown in Fig. 3 together with monthly precipitation. The maximum values of irrigation are estimated in the month of June, as expected from the analysis of soil moisture time series (Fig. 2). For some years, 2019 and 2022, irrigation starts earlier already in April and May whereas for the year 2016 irrigation occurred mostly in July and August. The estimated irrigation in March and October might be due to errors or to real irrigation signals in case of late-spring or winter crops, and it deserves to be further investigated by looking at ground information (not currently available to the authors).

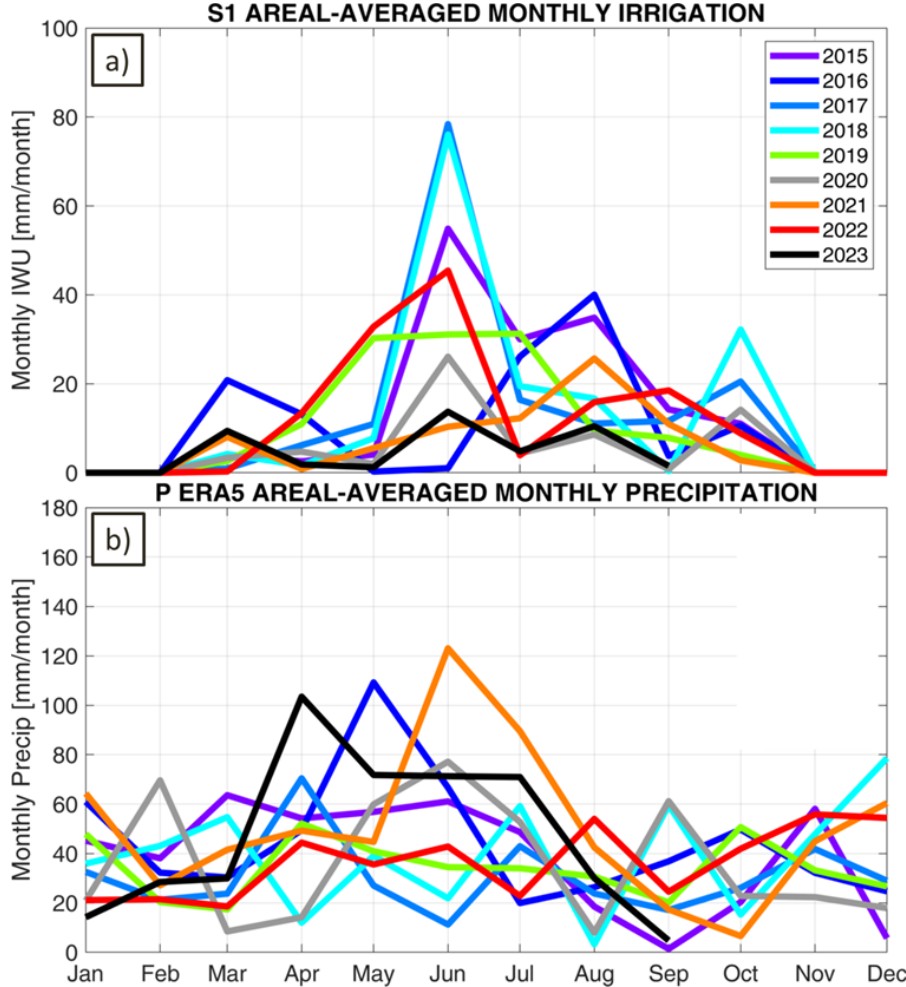

**Figure 3: Monthly values from 2015 to 2023 for the selected irrigated region for: a) irrigation water use, and b) precipitation. The comparison of irrigation water use and precipitation data provides important insights, as, for instance, the very low irrigation in 2023 notwithstanding the high precipitation in April and May.**

165

The annual values of irrigation water use are shown in Fig. 4 together with the map of annual irrigation averaged over the 9-year of the analysis; the yearly maps of irrigation water use are shown in Fig. A3 of the Appendix. As expected, in 2023 the irrigation water use was the lowest due to the Kakhovka dam's collapse with a mean value equal to 43 mm (+/- 55 mm), much lower than the average in the area equal to 115 mm (+/- 65 mm) (see the comparison of true-colour Sentinel-2 images

170 of 2017 and 2023 in Fig. A4 of the Appendix). Indeed, in 2023 the climatic conditions were good thanks to the large amount of precipitation in the months of April and May (Fig. 3b). In 2020, with 64 mm (+/- 42 mm), the second lowest value was recorded, probably due to the difficulties introduced by the COVID-19 pandemic, but also due to very low precipitation in the months of March and April that are crucial to store water for irrigation (Fig. 3b). We should note that the low availability of Sentinel-1 data in July 2020 might have also impacted the results. In 2017 and 2018 the maximum values were obtained

175 equal to 156 mm (+/- 64 mm) and 158 mm (+/- 67 mm), respectively.

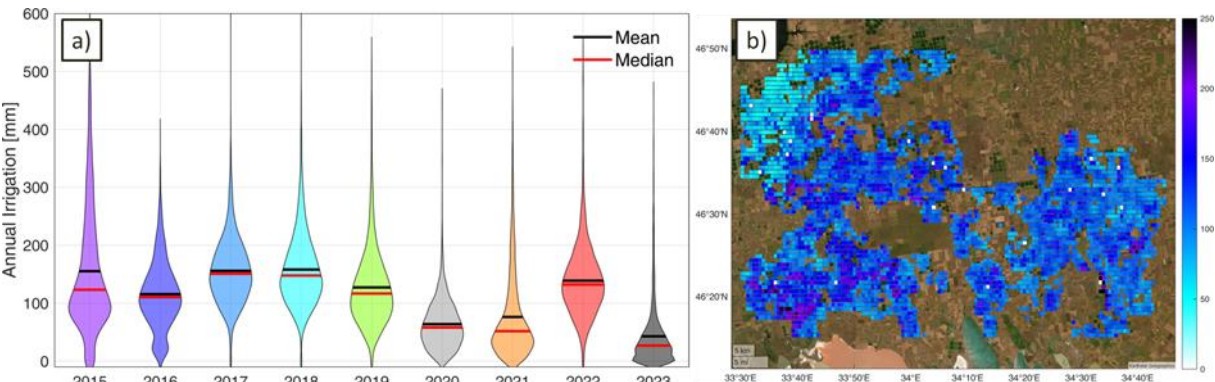

**Figure 4: a) Violin plot of the annual values of irrigation water use from 2015 to 2023, and b) mean annual irrigation water use map (background satellite image from Earthstar Geographics). In 2023, due to the Kakhovka dam's collapse, the lowest value was obtained.**

## 5 Conclusions

This study presents an innovative method for operational irrigation water use estimation without relying on ground observations. The method can be implemented in any location with accessible soil moisture, evaporation, and precipitation data. The soil moisture product is the most pertinent data set, and the suitability of Sentinel-1 satellite(s) to accurately ascertain irrigation water use has clearly been demonstrated owing to the high temporal and spatial resolution offered. The current temporal coverage of Sentinel-1-derived observations may be a limitation, but the continuity foreseen for the mission will offer the possibility of creating long-term time series of irrigation water use in the upcoming years.

The newly developed irrigation water use data set offer important insights on the socio-political dynamics in Ukraine. Indeed, the impact of Kakhovka dam's collapse was dramatic for the possibility of irrigating the agricultural fields in 2023. Notwithstanding the good amount of precipitation in April and May, the dam's collapse prevented the use of water for irrigation, with significant impacts on crop production. The possible impact of COVID-19 pandemic is also highlighted.

Nowadays, advancements in high-resolution satellite technology have provided an opportunity to monitor the environment with greater accuracy. This has led to the creation of new products which are essential for assessing the impact of natural and human disasters on the socio-political dynamics of our Earth system.

**Author contribution**

JD: Methodology, Software, Validation, Investigation, Writing – original draft preparation, Writing - review & editing. PF: Resources, Methodology, Investigation, Validation, Writing – review & editing. LB: Conceptualization, Methodology, Software, Validation, Formal analysis, Investigation, Writing – original draft preparation, Supervision, Visualization.

**Competing interest**

The authors declare that they have no conflict of interest.

**Acknowledgements**

The authors acknowledge support from the European Space Agency (ESA) projects IRRIGATION+ (contract number 4000129870/20/I-NB) and 4DMED-Hydrology (4000136272/21/I-EF).

 **Appendix**

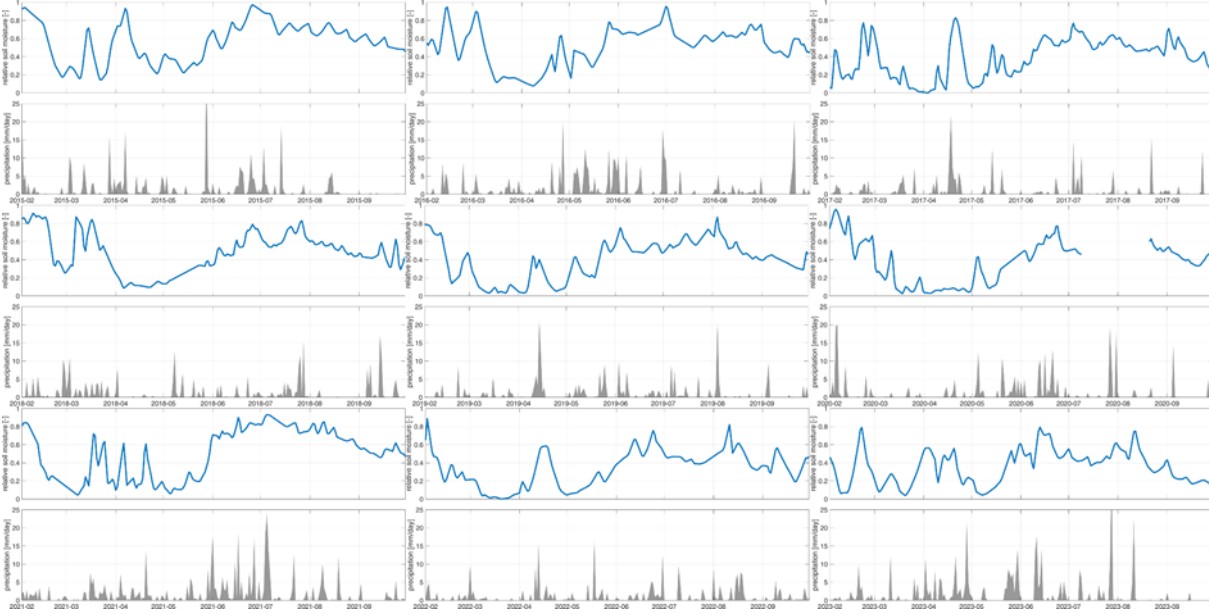

**Figure A1: Time series of spatially averaged relative soil moisture from Sentinel-1 CGLS product (upper panels) and precipitation from ERA5 Land (lower panels) during the whole study period. The spatial average is computed for pixels with average soil moisture greater than 45%.**


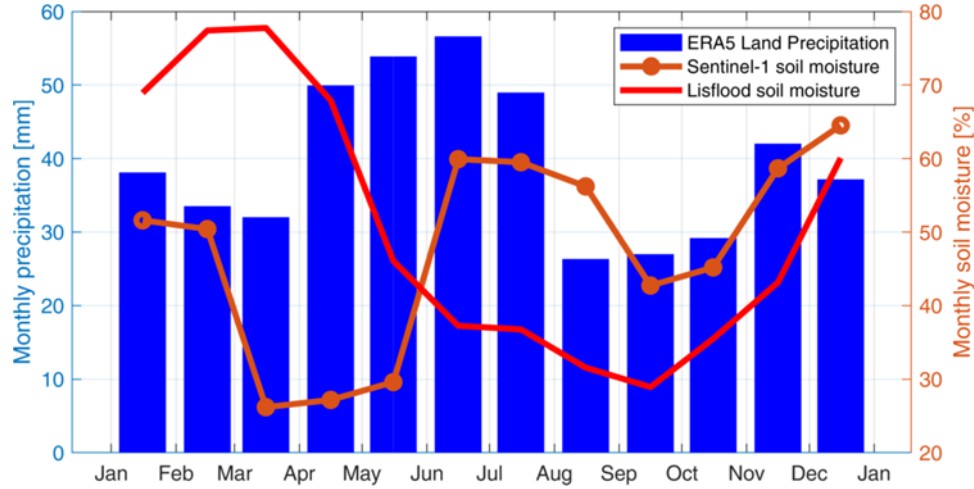

**Figure A2: Climatology of precipitation from ERA5 Land, satellite soil moisture from Sentinel-1 CGLS product and modelled soil moisture from Lisflood.**

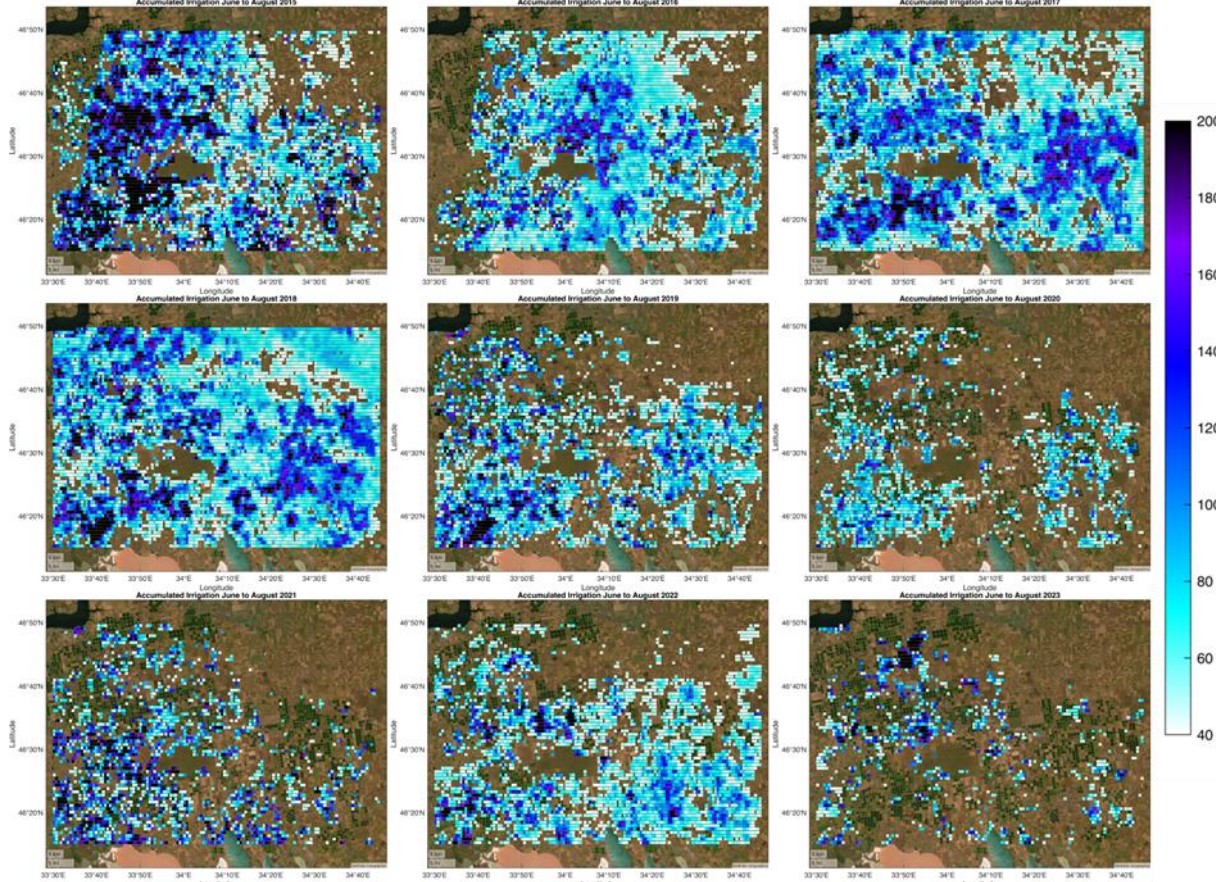

**215** **Figure A3: Accumulated irrigation in the period from June to August of each year, from 2015 to 2023, as obtained by the SM-based inversion approach (background satellite image from Earthstar Geographics).**

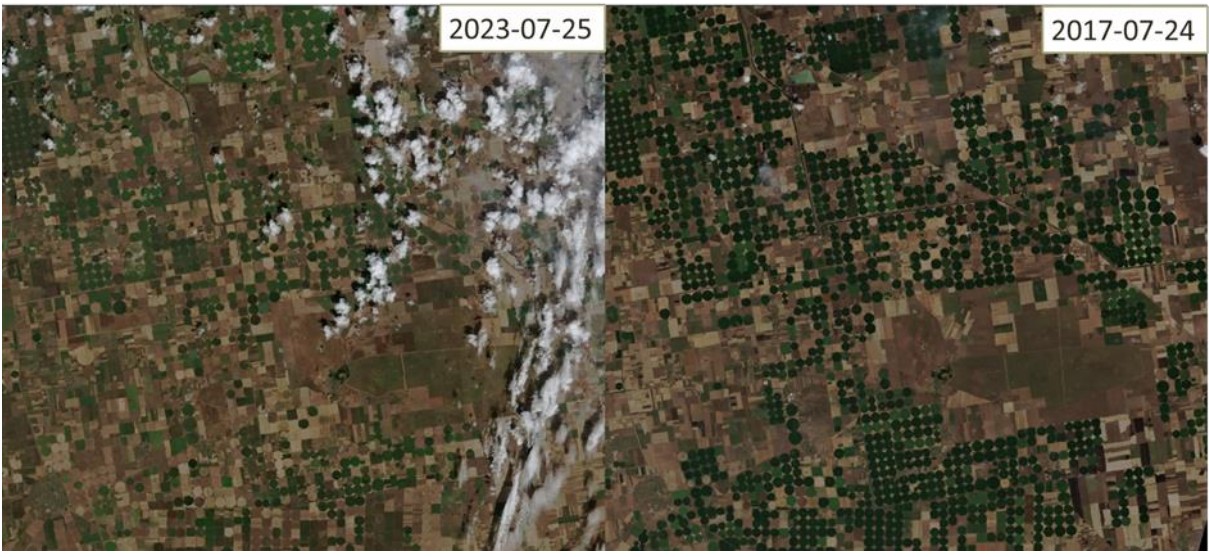

**220** **Figure A4: Comparison of true-colour Sentinel-2 images of 2017 and 2023 (obtained from Sentinel Hub EO Browser service).**

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
