# Peer review of "The development of an operational system for estimating irrigation water use reveals socio-political dynamics in Ukraine"

_EGUsphere, 2023_

## Author Comment (AC1)

**Response to the reviewer's comments**

The authors of this study wish to thank the reviewer for the useful comments.

In black, we have reported the reviewer's comments, in red, the detailed replies, and in blue, the sentences that will be changed and/or added in the revised manuscript to address the reviewer's comments.

R1C1. The manuscript titled "The development of an operational system for estimating irrigation water use reveals socio-political dynamics in Ukraine" introduces a system that leverages remote sensing data to estimate irrigation water use. This system is notably valuable for providing crucial data in water resource management, such as for modeling and monitoring. The application of this system is demonstrated using a timely case study of the Russian-Ukraine conflict. However, while the system shows great promise, the manuscript falls short in certain areas:

- The background research that led to the development of the system is inadequately detailed, particularly regarding the estimation of irrigation water use.
- The methodology section is too concise, lacking a comprehensive introduction or synthesis (if some methods are previously published) of the key methods or algorithms employed in this system.

R1A1. We are glad that the reviewer recognizes the significance of the work and we appreciate the comments provided for improving the manuscript quality. One main concern raised by the reviewer is the lack of background on the methodology adopted for estimating the irrigation water use, i.e., the SM-based inversion approach. This was our explicit intention, as the method is currently well-established after several development steps described in the following papers: Brocca et al. (2018), Dari et al. (2020, 2022, 2023). In this case, we decided to demonstrate our ability to implement an operational application that deals with a timely topic. Note that the reader is referred to the papers mentioned above for details on the method (see, e.g., lines 29, 34-41, 57, 69 or the sentence "the details for the implementation of the method are fully described in Dari et al. (2023)" at lines 72-73. Nevertheless, in order to meet the reviewer requirements, we will expand more on the method background in Section 3. The new Materials and Methods section will read as follows (added parts are underlined):

**"3 Materials and Methods**

**3.1 The SM-based inversion approach**

Irrigation water use has been estimated through the SM-based inversion approach (Brocca et a., 2018; Dari et al., 2020; 2023) over a time span ranging from January 1, 2015 to September 30, 2023. The core idea behind the method is the inversion of the soil water balance for backwards estimating the total water input, generally represented by rainfall plus irrigation. By expressing the soil water balance as:

$$Z^* \frac{dS(t)}{dt} = i(t) + r(t) - g(t) - sr(t) - e(t) \tag{1}$$

where $Z^*$ [mm] is the water capacity of the soil layer, $S(t)$ [-] is the relative soil moisture (i.e., ranging between 0 and 1), $t$ [days] indicates the time, $i(t)$ is the irrigation rate [mm/day], $r(t)$ [mm/day] is the rainfall rate, $g(t)$ [mm/day] is the drainage term, $sr(t)$ [mm/day] is the surface runoff, and $e(t)$ [mm/day] is the evapotranspiration rate. Eq. (1) is equivalent to:

$$Win(t) = Z^* \frac{dS(t)}{dt} + g(t) + sr(t) + e(t) \tag{2}$$

where $Win(t)$ is the total amount of water entering into the soil, i.e., rainfall plus irrigation. As thoroughly explained in previous studies by the authors, the following assumptions can be adopted: (i) $g(t) = aS(t)^b$ (Brocca et al., 2014), (ii) $sr(t) = 0$ (Brocca et al., 2015), (iii) $e(t) = F \cdot S(t) \cdot PET(t)$ (Dari et al., 2023). Hence, Eq. (2) can be rewritten as:

$$Win\ (t) = Z^* \frac{dS(t)}{dt} + aS(t)^b + F \cdot S(t) \cdot PET(t) \tag{3}$$

After estimating the total amount of water entering the soil, irrigation rates can be derived by removing rainfall rates from the total, $i(t) = Win\ (t) - r(t)$. Negative irrigation rates, if any, are imposed equal to zero (Jalilvand et al., 2019). A threshold value for the ratio between weekly estimated irrigation and weekly rainfall equal to 0.2 is adopted to discard negligible irrigation amounts due to random errors.

The parameters $a, b, Z^*$, and $F$ of Eq. (3) are the model parameters. $a, b$, and $Z^*$ have been calibrated against rainfall (i.e., by optimizing the method performances in properly reproducing rainfall amounts) by masking out days with no rainfall rate during the irrigation seasons (hence, potential irrigation days). The F parameter has been set equal to 0.3 as explained in Section 3.2. For further details on the method, the reader is referred to Dari et al. (2023).

**3.2 Data and processing**

The algorithm requires soil moisture, rainfall and potential evapotranspiration (PET) data as an input. We have used Sentinel-1 surface soil moisture observations from the Copernicus Global Land Service (https://land.copernicus.eu/global/products/ssm) (Bauer-Marschallinger et al., 2019) having a spatial resolution of 1 km and 2 to 6 days revisit time depending on the region of interest and the number of satellites available in orbit (2 satellites from October 2016 to December 2021, and 1 satellite from October 2014 to October 2016 and from January 2022 to September 2023 due to failure of Sentinel-1B). Before running the algorithm, the noise in the soil moisture signal has been reduced by computing the Soil Water Index (SWI) according to the exponential filter proposed by Albergel et al. (2008). Precipitation and PET have been obtained from the 5th land reanalysis of the European Centre for Medium-Range Weather Forecasts (ERA5 Land, https://cds.climate.copernicus.eu/cdsapp#!/dataset/reanalysis-era5-land?tab=overview), characterised by a native spatial resolution of 9 km and hourly temporal resolution (Muñoz-Sabater et al., 2021). As PET from ERA5 Land represents pan evaporation, i.e., open water evaporation, we have rescaled the data, i.e., to obtain the same mean value in the common period, by using potential evaporation from the Global Land Evaporation Amsterdam Model (GLEAM v3.6a, https://www.gleam.eu/) (Miralles et al., 2011; Martens et al., 2017) that was used in previous studies (e.g., Dari et al., 2023) but it is not available with short latency. The applied scaling value is equal to 30%. Precipitation and PET data have been resampled over the same spatial grid of Sentinel-1 with the nearest neighbouring approach and aggregated at daily time scale. Therefore, irrigation water use estimates have been produced on the 1 km grid of Sentinel-1 surface soil moisture data with a temporal resolution of 15 days. "

R1C2. I recognize the potential of the system proposed, but believe the manuscript requires substantial improvement to effectively showcase this important work. Given the extent of revisions needed, I recommend rejecting its current version. However, I strongly encourage resubmission of a revised version. I provided detailed thoughts and suggestions below which might guide these improvements.

First, the introduction section of the manuscript currently offers a limited synthesis of the research context. While there is an abundance of references to studies in remote sensing algorithms, the manuscript fails to adequately address the background in estimating irrigation water use. Given that the primary objective of

your system is to estimate and monitor irrigation water use, the absence of a thorough discussion on the research gaps in existing systems for estimating irrigation water use significantly weakens the presentation of your proposed system. To enhance this aspect, I recommend the following revisions for the introduction: (1) incorporate a comprehensive literature review that emphasizes the significance of estimating irrigation water use in water resource management and identifies existing research gaps; (2) establish a clear connection between these identified research gaps and your proposed system, highlighting how your system aims to bridge these gaps.

R1A2. We will revise the Introduction according to the reviewer's suggestions. We remark that estimating irrigation water use from satellite data is in itself an open challenge, but here, based on the authors previous works, we set ourselves an even more ambitious goal, which is the building of an operational system. Under this perspective, we balanced the introduction in order to not put too much focus on details of other methods to estimate irrigation water use available in literature (which are mentioned anyway), but emphasising the background of the SM-based inversion approach, here used to develop an operational system. The importance of estimating irrigation water use in water resource management and the existing research gaps (i.e., the total absence of operational systems) will be further emphasised. The new Introduction will read as follows (added parts are underlined):

**"1 Introduction**

In recent years, Europe has experienced a number of catastrophic events, including the COVID-19 pandemic and the conflict in Ukraine. The effects of these events are far-reaching and impact all sectors of society, including agriculture and food production (Van Tricht et al., 2023). Ukraine is among the largest wheat producers in Europe, and is indeed known as the breadbasket of Europe. It is crucial to comprehend the impact of these catastrophic events on crop production in Ukraine, particularly in the conflict-affected Kherson region. Crop production in the Kherson area south of the Kakhovka dam heavily depends on irrigation that is facilitated by the Kakhovka reservoir. As the Kakhovka dam's collapse on June 6, 2023 is anticipated to have had a significant impact on crop production, it could be evaluated by examining the variability over time of irrigation water use in the area.

Thanks to the advance in satellite technology, as for instance the launch of the Sentinel constellation under the Copernicus Programme, remote sensing has recently enabled the acquisition of irrigation water use measurements (Massari et al., 2021; McDermid et al., 2023), enabling large areas to be monitored in a consistent and equitable manner. This circumstance opens unprecedented perspectives in water resources management over human-altered basins. In fact, irrigation represents the largest component of the anthropogenic water use (Foley et al., 2011; Dorigo et al., 2021), with impacts on several components of the Earth system and social dynamics (McDermid et al., 2023; Dari et al., 2024). In general, satellite observations of hydrological variables that can be a proxy of irrigation occurrence are used to estimate irrigation volumes, as long as the condition of a matching between the spatio-temporal resolution of the observational data and the spatial and temporal scales of irrigation dynamics is satisfied (Dari et al., 2022; Zappa et al., 2022). Specifically, approaches based on satellite soil moisture (e.g., Lawston et al., 2017; Brocca et al., 2018; Dari et al., 2020) and evaporation (e.g., Bretreger et al., 2022; Brombacher et al., 2022; Kragh et al., 2023) products have been developed in recent years. An example integrating both soil moisture and evaporation products is the soil moisture based (SM-based) inversion approach developed by Dari et al. (2023) as an evolution of the SM2RAIN (Soil Moisture to RAINfall) algorithm originally designed to estimate rainfall from satellite soil moisture (Brocca et al., 2014). Preliminary promising results were shown by Brocca et al. (2018) and Filippucci et al. (2020) by means of coarse resolution satellite and in-situ soil moisture, respectively. Concurrently, a few studies deepened the role of the evapotranspiration term within the algorithm structure (Jalilvand et al., 2019; Dari et al., 2020; 2022b). The first implementation with high-resolution satellite soil moisture as an

input has been proposed by Dari et al. (2020). The authors produced a data set of irrigation estimates at 1 km spatial resolution over a heavily irrigated portion of the Ebro basin, in Spain, covering the period 2011-2017. Recently, the SM-based inversion approach has been implemented under the European Space Agency (ESA) Irrigation+ project for producing the first regional-scale, high-resolution data sets over three major basins worldwide (Dari et al., 2023). In a nutshell, the SM-based inversion approach proved itself to be a useful tool for estimating irrigation water use across scales; the following natural step is the exploration of the possibility of building an operational system based on it, as currently no operational services for monitoring large-scale irrigation are available.

In this study, we have developed for the first time an operational system for monitoring irrigation water use with 10 days latency relying on the SM-based inversion approach forced with operational satellite-based surface soil moisture data and precipitation and evaporation data from reanalysis. "

R1C3. Second, Section 3 on Materials and Methods currently provides insufficient information for readers. The section appears to predominantly describe the necessary data and data sources, yet it neglects to adequately detail the critical methods or algorithms that underpin the system. Consequently, while the data requirements for the system are clear, its structure (such as the system components, modules, and data flows) and the core algorithms that facilitate its operation remain obscure. Additionally, the results and discussion sections indicate the application of the system to a real-world scenario for demonstration purposes, yet the method section lacks a comprehensive overview of the chosen case area. I recommend that the authors thoroughly reevaluate and substantially revise the methodology section to provide a clearer and more detailed presentation of the proposed system. This revision should aim to explain in detail for both the system architecture and its foundational algorithms, as well as incorporate a more detailed description of the case study area within the methodological framework.

R1A3. We believe that we already responded about changes to be done to better describe the methodology in R1A1. We will also improve the description of the study site, adding more details on the irrigation infrastructure of the Kakhova system. The revised version of the Study Area section will read as follows (added parts are underlined):

"2 Study area

As a proof of concept, the operational system for monitoring irrigation water use from satellite data has been implemented over a cold semi-arid area (Beck et al., 2018) enclosing a heavily irrigated portion fed by the Kakhovka reservoir on the Dnipro river, in Ukraine, collapsed on June 6, 2023. More in detail, we have selected a box of almost 4000 km2 whose extension ranges from longitude 33.30° to 34.45° and from latitude 46.15° to 46.50°. This is the area fed by the Kakhovsky canal, which originates just upstream the dam and delivers water to five irrigation districts through an efficient and automated network; the districts are equipped with a dense system of centre pivot that was mainly realized between the late 1970s and 1980s as part of the development of the Kakhova system, completed in 1990 (Kuns, 2018) and representing one of the largest irrigated areas in Europe. The dense system of center pivot irrigation equipment can be observed by visual inspection of satellite maps (see, e.g., Fig. 1a). For the selected area, the latest version of Global Map of Irrigated Areas (GMIA) (Mehta et al., 2022) reports peaks up to 60% in terms of percentage of area equipped for irrigation. The data set refers to cells characterised by a spatial resolution of 5 arc-minutes (about 10 km at the Equator). Reznik et al. (2016) report a percentage of irrigated areas equal to 83.3% of the total available area in 2015. Based on statistical surveys, the main cropping season for cereal and other annual crops in Ukraine is from May to August (Portmann et al., 2008)."

R1C4. I would like to reiterate that the topic and the proposed system are both valuable and of great interest to me. However, the current presentation of the manuscript does not do justice to the value of the work. It is essential that the content and delivery are refined to effectively convey the significance of your research. I look forward to having the opportunity to review a revised and resubmitted version of this manuscript in the future, hoping it will fully reveal the potential and importance of the work.

R1A4. We thank the reviewer.

**References**

Brocca, L., Tarpanelli, A., Filippucci, P., Dorigo, W., Zaussinger, F., Gruber, A. and Fernández-Prieto, D.: How much water is used for irrigation? A new approach exploiting coarse resolution satellite soil moisture products. International Journal of Applied Earth Observation and Geoinformation, 73C, 752-766, doi:10.1016/j.jag.2018.08.023, 2018.

Dari, J., Brocca, L., Quintana-Seguí, P., Escorihuela, M.J., Stefan, V. and Morbidelli, R.: Exploiting high-resolution remote sensing soil moisture to estimate irrigation water amounts over a Mediterranean region. Remote Sensing, 12(16), 2593, doi:10.3390/rs12162593, 2020.

Dari, J., Quintana-Seguí, P., Morbidelli, R., Saltalippi, C., Flammini, A., Giugliarelli, E., Escorihuela, M.J., Stefan, V. and Brocca, L.: Irrigation estimates from space: implementation of different approaches to model the evapotranspiration contribution within a soil-moisture-based inversion algorithm. Agricultural Water Management, 265, 107537, doi:10.1016/j.agwat.2022.107537, 2022.

Dari, J., Brocca, L., Modanesi, S., Massari, C., Tarpanelli, A., Barbetta, S., Quast, R., Vreugdenhil, M., Freeman, V., Barella-Ortiz, A., Quintana-Seguí, P., Bretreger, D. and Volden, E.: Regional data sets of high-resolution (1 and 6 km) irrigation estimates from space. Earth System Science Data, 15(4), 1555-1575, doi:10.5194/essd-15-1555-2023, 2023.

---

## Author Comment (AC2)

**Response to the reviewer's comments**

The authors of this study wish to thank the reviewer for the useful comments.

In black, we have reported the reviewer's comments, in red, the detailed replies, and in blue, the sentences that will be changed and/or added in the revised manuscript to address the reviewer's comments.

R2C1. This study used data from satellite soil moisture, reanalysis precipitation and potential evaporation to estimate irrigation water use. The topic of this study is in general interesting. However, there are some issues that need to be addressed with the connection to introduction, materials and methods, discussions, and potential uncertainties or caveats of the results from this study. I feel that these points need substantial improvements, and, thus, I would recommend rejecting its current form.

R2A1. We thank the reviewer for the useful suggestions provided. Please find a detailed response as follows.

**General comments:**

**Introduction**:

R2C2. The background and necessity of conducting this research are not adequately introduced. It is very different for readers to understand the how this topic has been developed in the community, and what is the innovation of the current study. I do not think that „no operational services for monitoring large-scale irrigation are available" can be the research gap for the scientific journal of HESS.

R2A2. We will revise the Introduction section, considering suggestions from the other reviewer as well. We remark that estimating irrigation water use from satellite data is in itself an open challenge, but here, based on the authors previous works, we set ourselves an even more ambitious goal, which is the building of an operational system. Under this perspective, we balanced the introduction in order to not put too much focus on details of other methods to estimate irrigation water use available in literature (which are mentioned anyway), but emphasising the background of the SM-based inversion approach, here used to develop an operational system. Regarding this point, we disagree with the reviewer's point of view, as we believe instead that the possibility of building a satellite-based operational system for monitoring agricultural water use is a primary scientific challenge and of paramount importance for associated societal implications. Irrigation is, in fact, concurrently the most unknown but predominant human alteration on the water cycle. Hence, the possibility of estimating irrigation water use from remote sensing is essential for properly closing the water budget over anthropized basins. We believe that the perspective of bringing this research challenge into an operational framework can be of wide interest for the HESS journal, given the important implications on water resources management. The interest of the hydrological scientific community in this research line is also affirmed by the recent study by Manivasagam (2024). Finally, we remark the difference with previous studies from the authors, which were instead aimed at developing the proposed approach (see, e.g., Brocca et al., 2018; Dari et al., 2020; 2023).

The revised version of the Introduction will read as follow (added parts are underlined):

**"1 Introduction**

In recent years, Europe has experienced a number of catastrophic events, including the COVID-19 pandemic and the conflict in Ukraine. The effects of these events are far-reaching and impact all sectors of society, including agriculture and food production (Van Tricht et al., 2023). Ukraine is among the largest wheat producers in Europe, and is indeed known as the breadbasket of Europe. It is crucial to comprehend the impact of these catastrophic events on crop production in Ukraine, particularly in the conflict-affected

Kherson region. Crop production in the Kherson area south of the Kakhovka dam heavily depends on irrigation that is facilitated by the Kakhovka reservoir. As the Kakhovka dam's collapse on June 6, 2023 is anticipated to have had a significant impact on crop production, it could be evaluated by examining the variability over time of irrigation water use in the area.

Thanks to the advance in satellite technology, as for instance the launch of the Sentinel constellation under the Copernicus Programme, remote sensing has recently enabled the acquisition of irrigation water use measurements (Massari et al., 2021; McDermid et al., 2023), enabling large areas to be monitored in a consistent and equitable manner. This circumstance opens unprecedented perspectives in water resources management over human-altered basins. In fact, irrigation represents the largest component of the anthropogenic water use (Foley et al., 2011; Dorigo et al., 2021), with impacts on several components of the Earth system and social dynamics (McDermid et al., 2023; Dari et al., 2024). In general, satellite observations of hydrological variables that can be a proxy of irrigation occurrence are used to estimate irrigation volumes, as long as the condition of a matching between the spatio-temporal resolution of the observational data and the spatial and temporal scales of irrigation dynamics is satisfied (Dari et al., 2022; Zappa et al., 2022). Specifically, approaches based on satellite soil moisture (e.g., Lawston et al., 2017; Brocca et al., 2018; Dari et al., 2020) and evaporation (e.g., Bretreger et al., 2022; Brombacher et al., 2022; Kragh et al., 2023) products have been developed in recent years. An example integrating both soil moisture and evaporation products is the soil moisture based (SM-based) inversion approach developed by Dari et al. (2023) as an evolution of the SM2RAIN (Soil Moisture to RAINfall) algorithm originally designed to estimate rainfall from satellite soil moisture (Brocca et al., 2014). Preliminary promising results were shown by Brocca et al. (2018) and Filippucci et al. (2020) by means of coarse resolution satellite and in-situ soil moisture, respectively. Concurrently, a few studies deepened the role of the evapotranspiration term within the algorithm structure (Jalilvand et al., 2019; Dari et al., 2020; 2022b). The first implementation with high-resolution satellite soil moisture as an input has been proposed by Dari et al. (2020). The authors produced a data set of irrigation estimates at 1 km spatial resolution over a heavily irrigated portion of the Ebro basin, in Spain, covering the period 2011-2017. Recently, the SM-based inversion approach has been implemented under the European Space Agency (ESA) Irrigation+ project for producing the first regional-scale, high-resolution data sets over three major basins worldwide (Dari et al., 2023). In a nutshell, the SM-based inversion approach proved itself to be a useful tool for estimating irrigation water use across scales; the following natural step is the exploration of the possibility of building an operational system based on it, as currently no operational services for monitoring large-scale irrigation are available.

In this study, we have developed for the first time an operational system for monitoring irrigation water use with 10 days latency relying on the SM-based inversion approach forced with operational satellite-based surface soil moisture data and precipitation and evaporation data from reanalysis. "

R2C3. **Study area:** There are lack of enough information about the study area, for example, climate, agriculture, or societal and political situation.

R2A3. We will revise the Study area section according to the reviewer's suggestions. The climatic context is mentioned at line 48 of the manuscript. We will add more information on the irrigation system installed there, which is of interest with respect to the aim of the paper. The revised version will read as follows (added parts are underlined):

**"2 Study area**

As a proof of concept, the operational system for monitoring irrigation water use from satellite data has been implemented over a cold semi-arid area (Beck et al., 2018) enclosing a heavily irrigated portion fed by the

Kakhovka reservoir on the Dnipro river, in Ukraine, collapsed on June 6, 2023. More in detail, we have selected a box of almost 4000 km2 whose extension ranges from longitude 33.30° to 34.45° and from latitude 46.15° to 46.50°. This is the area fed by the Kakhovsky canal, which originates just upstream the dam and delivers water to five irrigation districts through an efficient and automated network; the districts are equipped with a dense system of centre pivot that was mainly realized between the late 1970s and 1980s as part of the development of the Kakhova system, completed in 1990 (Kuns, 2018) and representing one of the largest irrigated areas in Europe. The dense system of center pivot irrigation equipment can be observed by visual inspection of satellite maps (see, e.g., Fig. 1a). For the selected area, the latest version of Global Map of Irrigated Areas (GMIA) (Mehta et al., 2022) reports peaks up to 60% in terms of percentage of area equipped for irrigation. The data set refers to cells characterised by a spatial resolution of 5 arc-minutes (about 10 km at the Equator). Reznik et al. (2016) report a percentage of irrigated areas equal to 83.3% of the total available area in 2015. Based on statistical surveys, the main cropping season for cereal and other annual crops in Ukraine is from May to August (Portmann et al., 2008)."

R2C4. **Materials and Methods**: A detailed description of the SM-based inversion approach is need in this section for readability.

R2A4. The Materials and Methods section will be re-organized and improved. A detailed description of the method will be added. The revised version of the section will read as follows (added parts are underlined):

**"3 Materials and Methods**

**3.1 The SM-based inversion approach**

Irrigation water use has been estimated through the SM-based inversion approach (Brocca et a., 2018; Dari et al., 2020; 2023) over a time span ranging from January 1, 2015 to September 30, 2023. The core idea behind the method is the inversion of the soil water balance for backwards estimating the total water input, generally represented by rainfall plus irrigation. By expressing the soil water balance as:

$$Z^* \frac{dS(t)}{dt} = i(t) + r(t) - g(t) - sr(t) - e(t) \tag{1}$$

where $Z^*$ [mm] is the water capacity of the soil layer, $S(t)$ [-] is the relative soil moisture (i.e., ranging between 0 and 1), $t$ [days] indicates the time, $i(t)$ is the irrigation rate [mm/day], $r(t)$ [mm/day] is the rainfall rate, $g(t)$ [mm/day] is the drainage term, $sr(t)$ [mm/day] is the surface runoff, and $e(t)$ [mm/day] is the evapotranspiration rate. Eq. (1) is equivalent to:

$$Win(t) = Z^* \frac{dS(t)}{dt} + g(t) + sr(t) + e(t) \tag{2}$$

where $Win(t)$ is the total amount of water entering into the soil, i.e., rainfall plus irrigation. As thoroughly explained in previous studies by the authors, the following assumptions can be adopted: (i) $g(t) = aS(t)^b$ (Brocca et al., 2014), (ii) $sr(t) = 0$ (Brocca et al., 2015), (iii) $e(t) = F \cdot S(t) \cdot PET(t)$ (Dari et al., 2023). Hence, Eq. (2) can be rewritten as:

$$Win(t) = Z^* \frac{dS(t)}{dt} + aS(t)^b + F \cdot S(t) \cdot PET(t) \tag{3}$$

After estimating the total amount of water entering the soil, irrigation rates can be derived by removing rainfall rates from the total, $i(t) = Win(t) - r(t)$. Negative irrigation rates, if any, are imposed equal to zero (Jalilvand et al., 2019). A threshold value for the ratio between weekly estimated irrigation and weekly rainfall equal to 0.2 is adopted to discard negligible irrigation amounts due to random errors.

The parameters $a$, $b$, $Z^*$, and $F$ of Eq. (3) are the model parameters. $a$, $b$, and $Z^*$ have been calibrated against rainfall (i.e., by optimizing the method performances in properly reproducing rainfall amounts) by masking out days with no rainfall rate during the irrigation seasons (hence, potential irrigation days). The F parameter has been set equal to 0.3 as explained in Section 3.2. For further details on the method, the reader is referred to Dari et al. (2023).

**3.2 Data and processing**

The algorithm requires soil moisture, rainfall and potential evapotranspiration (PET) data as an input. We have used Sentinel-1 surface soil moisture observations from the Copernicus Global Land Service (https://land.copernicus.eu/global/products/ssm) (Bauer-Marschallinger et al., 2019) having a spatial resolution of 1 km and 2 to 6 days revisit time depending on the region of interest and the number of satellites available in orbit (2 satellites from October 2016 to December 2021, and 1 satellite from October 2014 to October 2016 and from January 2022 to September 2023 due to failure of Sentinel-1B). Before running the algorithm, the noise in the soil moisture signal has been reduced by computing the Soil Water Index (SWI) according to the exponential filter proposed by Albergel et al. (2008). Precipitation and PET have been obtained from the 5th land reanalysis of the European Centre for Medium-Range Weather Forecasts (ERA5 Land, https://cds.climate.copernicus.eu/cdsapp#!/dataset/reanalysis-era5-land?tab=overview), characterised by a native spatial resolution of 9 km and hourly temporal resolution (Muñoz-Sabater et al., 2021). As PET from ERA5 Land represents pan evaporation, i.e., open water evaporation, we have rescaled the data, i.e., to obtain the same mean value in the common period, by using potential evaporation from the Global Land Evaporation Amsterdam Model (GLEAM v3.6a, https://www.gleam.eu/) (Miralles et al., 2011; Martens et al., 2017) that was used in previous studies (e.g., Dari et al., 2023) but it is not available with short latency. The applied scaling value is equal to 30%. Precipitation and PET data have been resampled over the same spatial grid of Sentinel-1 with the nearest neighbouring approach and aggregated at daily time scale. Therefore, irrigation water use estimates have been produced on the 1 km grid of Sentinel-1 surface soil moisture data with a temporal resolution of 15 days. "

R2C5. Page 3, Line 70: Could the authors explain the meaning of "scaling value" and why 30% is used.

R2A5. We will add a sentence about the meaning of "scaling value". Such a sentence, together with the reason why 30% is adopted, can be found in the new "3.2 Data and processing" section (see R2A4). We remark here the rationale behind this choice. Since PET from ERA5 Land represents pan evaporation, i.e., open water evaporation, we have rescaled the data, i.e., to obtain the same mean value in the common period, by using potential evaporation from the GLEAM v3.6a that was used in previous studies (e.g., Dari et al., 2023) but it is not available with short latency. A scaling value equal to 30% ($F$ parameter equal to 0.3) has been found to be suitable to obtain the same mean value between the two considered data sets in the common period.

**Results and discussion:**

R2C6. Page 3, Line 75-80: I think that this paragraph would fit in the methods section.

R2A6. We will move this part to the Methods section.

R2C7. Page 3, Line 82: How many years are selected for deriving Figure 1?

R2A7. The figure shows the mean soil moisture considering all the irrigation seasons (June-August) during the whole considered period, 2015-2023. We will specify the considered years in the text.

R2C8. Page 6, Line 121: I would like to encourage the authors to validate your irrigation results derived from satellites against ground observations.

R2A8. We totally agree with the reviewer that it would be nice to validate the retrieved irrigation amounts. Unfortunately, the general lack of in-situ observation of irrigation quantities is concurrently the main driver and the main limitation of this research line. This is a well-known issue (see, e.g., Brocca et al., 2018; Dorigo et al., 2021; McDermid et al., 2023, only to cite a few). To the best of our knowledge, this area is not an exception. In fact, we contacted both Universities and local authorities looking for reference data without succeeding. However, we remark that the SM-based inversion algorithm has been recently validated over three main basins worldwide (Dari et al., 2023), also with climatic features similar to those of the case study proposed here. Finally, it is also important to highlight that the reliability of the retrieved irrigation amounts is consistent with precipitation dynamics.

R2C9. In general, I have not seen thoughtful discussion in this section.

R2A9. The perspective of the proposed paper is to showcase the feasibility of building an operational system for monitoring irrigation water use through the SM-based inversion approach. Scientific discussion on potential and limits of the method would be out of context, as they have been previously deepened in other studies by the authors (see, e.g., Brocca et al., 2018; Dari et al., 2020; 2022; 2023). We believe that the discussion provided here is enough to also address the previous point (see R2A8), i.e., it is aimed at showing the reliability of results. Under this perspective, the discussion has been oriented at evaluating satellite soil moisture against rainfall dynamics. If the reviewer thinks that some specific point would enrich the discussion section, we would be glad to consider suggestions.

**Conclusions:**

R2C10. I agree with the authors that advancements in high-resolution satellite technology and new high-resolution productions, particularly, irrigation water use, are needed. However, the analysis and discussion in this study are not adequately support for these conclusions.

R2A10. We hope that previous replies have addressed this point.

**References**

Brocca, L., Tarpanelli, A., Filippucci, P., Dorigo, W., Zaussinger, F., Gruber, A. and Fernández-Prieto, D.: How much water is used for irrigation? A new approach exploiting coarse resolution satellite soil moisture products. International Journal of Applied Earth Observation and Geoinformation, 73C, 752-766, doi:10.1016/j.jag.2018.08.023, 2018.

Dari, J., Brocca, L., Quintana-Seguí, P., Escorihuela, M.J., Stefan, V. and Morbidelli, R.: Exploiting high-resolution remote sensing soil moisture to estimate irrigation water amounts over a Mediterranean region. Remote Sensing, 12(16), 2593, doi:10.3390/rs12162593, 2020.

Dari, J., Brocca, L., Modanesi, S., Massari, C., Tarpanelli, A., Barbetta, S., Quast, R., Vreugdenhil, M., Freeman, V., Barella-Ortiz, A., Quintana-Seguí, P., Bretreger, D. and Volden, E.: Regional data sets of high-resolution (1 and 6 km) irrigation estimates from space. Earth System Science Data, 15(4), 1555-1575, doi:10.5194/essd-15-1555-2023, 2023.

Dari, J., Quintana-Seguí, P., Morbidelli, R., Saltalippi, C., Flammini, A., Giugliarelli, E., Escorihuela, M.J., Stefan, V., and Brocca, L.: Irrigation estimates from space: Implementation of different approaches to model the

evapotranspiration contribution within a soil-moisture-based inversion algorithm, Agric. Water Manag., 265, 107537, https://doi.org/10.1016/j.agwat.2022.107537, 2022.

Dorigo, W., Dietrich, S., Aires, F., Brocca, L., Carter, S., Cretaux, J.-F., Dunkerley, D., Enomoto, H., Forsberg, R., Güntner, A., Hegglin, M.I., Hollmann, R., Hurst, D.F., Johannessen, J.A., Kummerow, C., Lee, T., Luojus, K., Looser, U., Miralles, D.G., Pellet, V., Recknagel, T., Vargas, C.R., Schneider, U., Schoeneich, P., Schröder, M., Tapper, N., Vuglinsky, V., Wagner, W., Yu, L., Zappa, L., Zemp, M., and Aich, V.: Closing the water cycle from observations across scales: where do we stand?, Bull. Am. Meteorol. Soc., 102 (10), E1897-E1935, doi: 10.1175/BAMS-D-19-0316.1, 2021.

Manivasagam, V.S.: Remote sensing of irrigation: Research trends and the direction to next-generation agriculture through data-driven scientometric analysis. Water Security, 21, 100161, doi: 10.1016/j.wasec.2023.100161, 2024.

McDermid, S., Nocco, M., Lawston-Parker, P., et al.: Irrigation in the Earth system. Nature Reviews Earth & Environment, 4, 435–453, doi:10.1038/s43017-023-00438-5, 2023.

---

## Author Comment (AC3)

**Response to the reviewer's comments**

The authors of this study wish to thank the reviewer for the useful comments.

In black, we have reported the reviewer's comments, in red, the detailed replies, and in blue, the sentences that will be changed and/or added in the revised manuscript to address the reviewer's comments.

R3C1. In this work, the authors discuss the importance of building an operational system for monitoring water-use quantity for large-scale irrigation purposes (for example, during high-impact dam-destructions, big-scale pandemics or wars), through satellite images, with spatial resolution of 1 km, of soil-moisture, precipitation and potential evapotranspiration (through ERA5 reanalysis data and GLEA models), and with application south of Kakhova in Ukraine and during the period 2015-2023. Although the idea seems interesting, further analysis and validation are required to support it. Please see some major issues that I hope they can be of help to the authors:

R3A1. We are glad for the reviewer's positive feedback. Please find a detailed response as follows.

R3C2. 1) I would recommend the Abstract and Introduction focus on the idea of "developing an operational system for monitoring water-use quantity for large-scale irrigation purposes", and mention to the end something like that "to support this idea, we select for application the largest scale events of the last decade in Europe, which is the COVID and the conflict in Ukraine that led to the destruction of high-impact irrigation dam", since there seems to be much emotion and focus on the application area (that, although justified, it is not the purpose of this paper).

R3A2. We agree with the reviewer, the case study has been chosen because it allows us to depict timely socio-political dynamics, but the paper's main intent is to develop an operational system for monitoring water-use quantity for large-scale irrigation purposes. We will rephrase the abstract as follows:

"Irrigation is the main driver for crop production in many agricultural regions across the world. The estimation of irrigation water has the potential to enhance our comprehension of the Earth system, thus providing crucial data for food production.

In this study, we have created a unique operational system for estimating irrigation water using data from satellite soil moisture, reanalysis precipitation and potential evaporation. As a proof of concept, we implemented the method at high-resolution (1 km) during the period of 2015-2023 over the area south of the Kakhovka dam in Ukraine, which collapsed on June 6, 2023. The selected study area enabled us to showcase that our operational system is able to track the effect of the pandemic and conflict on the irrigation water supply. A significant decrease of 63% and 44% in irrigation water compared to the mean irrigation water between 2015-2023 has been identified as being linked to the collapse of the dam and, potentially, to the COVID-19 pandemic, respectively."

R3C3. 2) I do not comprehend the sentence "Concurrently, a few studies deepened the role of the evapotranspiration term within the algorithm structure"; please consider further explaining it or maybe rephrasing it, and present what exactly these other studies' advantages and disadvantages.

R3A3. We will change the sentence as follows:

"Concurrently, a few studies demonstrated the importance of considering the evapotranspiration term within the algorithm structure together with SM (Jalilvand et al., 2019; Dari et al., 2020; 2022b)."

In addition, we believe that details on the methodology added to address R3C5 will further clarify the sentence in the context of the model's structure (see R3A5).

R3C4. 3) Please add more information for the study area (for example, why the dam collapsed, dimensions of the dam and its reservoir, temperature, PET, precipitation, soil-moisture, streamflow, etc.), so that the readers are familiarized with the area's social, climatic and hydrological conditions.

R3A4. The climatic context is mentioned at line 48 of the manuscript. We will add more information on the study area. The revised version will read as follows (added parts are underlined):

As a proof of concept, the operational system for monitoring irrigation water use from satellite data has been implemented over a cold semi-arid area (Beck et al., 2018) enclosing a heavily irrigated portion fed by the Kakhovka reservoir on the Dnipro river (approximate length of 2200 km and average flow at the outlet of 53 km$^3$/year under natural conditions) in Ukraine, collapsed on June 6, 2023. Under operating conditions, the store capacity was of 18.2 km$^3$, corresponding to an extent of water surface equal to 2155 km$^2$. We have selected a box of almost 4000 km$^2$ whose extension ranges from longitude 33.30° to 34.45° and from latitude 46.15° to 46.50°. This is the area fed by the Kakhovsky canal, which originates just upstream the dam and delivers water to five irrigation districts through an efficient and automated network; the districts are equipped with a dense system of centre pivot that was mainly realized between the late 1970s and 1980s as part of the development of the Kakhova system, completed in 1990 (Kuns, 2018) and representing one of the largest irrigated areas in Europe. The dense system of center pivot irrigation equipment can be observed by visual inspection of satellite maps (see, e.g., Fig. 1a). For the selected area, the latest version of Global Map of Irrigated Areas (GMIA) (Mehta et al., 2022) reports peaks up to 60% in terms of percentage of area equipped for irrigation. The data set refers to cells characterised by a spatial resolution of 5 arc-minutes (about 10 km at the Equator). Reznik et al. (2016) report a percentage of irrigated areas equal to 83.3% of the total available area in 2015. Based on statistical surveys, the main cropping season for cereal and other annual crops in Ukraine is from May to August (Portmann et al., 2008).

R3C5. 4) It is mentioned that "To assess irrigation water use from satellite observations of soil moisture (or evaporation), the observations must detect the increase in soil water associated with irrigation application."; however, it is not clear what exactly the methodology is, and how evaporation can be used instead of the soil-moisture. Please consider presenting the methodology in further detail and how exactly one can estimate soil-moisture or evaporation from the satellite image processing.

R3A5. In our method soil moisture is used together with evaporation. We hope this will be clarified by a detailed description of the method that will be added. The revised version of the section will read as follows (added parts are underlined):

**"3 Materials and Methods**

**3.1 The SM-based inversion approach**

Irrigation water use has been estimated through the SM-based inversion approach (Brocca et a., 2018; Dari et al., 2020; 2023) over a time span ranging from January 1, 2015 to September 30, 2023. The core idea behind the method is the inversion of the soil water balance for backwards estimating the total water input, generally represented by rainfall plus irrigation. By expressing the soil water balance as:

$$Z^* \frac{dS(t)}{dt} = i(t) + r(t) - g(t) - sr(t) - e(t) \qquad (1)$$

where $Z^*$ [mm] is the water capacity of the soil layer, $S(t)$ [-] is the relative soil moisture (i.e., ranging between 0 and 1), $t$ [days] indicates the time, $i(t)$ is the irrigation rate [mm/day], $r(t)$ [mm/day] is the rainfall rate, $g(t)$ [mm/day] is the drainage term, $sr(t)$ [mm/day] is the surface runoff, and $e(t)$ [mm/day] is the evapotranspiration rate. Eq. (1) is equivalent to:

$$Win(t) = Z^* \frac{dS(t)}{dt} + g(t) + sr(t) + e(t) \qquad (2)$$

where $Win(t)$ is the total amount of water entering into the soil, i.e., rainfall plus irrigation. As thoroughly explained in previous studies by the authors, the following assumptions can be adopted: (i) $g(t) = aS(t)^b$ (Brocca et al., 2014), (ii) $sr(t) = 0$ (Brocca et al., 2015), (iii) $e(t) = F \cdot S(t) \cdot PET(t)$ (Dari et al., 2023). Hence, Eq. (2) can be rewritten as:

$$Win(t) = Z^* \frac{dS(t)}{dt} + aS(t)^b + F \cdot S(t) \cdot PET(t) \qquad (3)$$

After estimating the total amount of water entering the soil, irrigation rates can be derived by removing rainfall rates from the total, $i(t) = Win(t) - r(t)$. Negative irrigation rates, if any, are imposed equal to zero (Jalilvand et al., 2019). A threshold value for the ratio between weekly estimated irrigation and weekly rainfall equal to 0.2 is adopted to discard negligible irrigation amounts due to random errors.

The parameters $a, b, Z^*$, and $F$ of Eq. (3) are the model parameters. $a, b$, and $Z^*$ have been calibrated against rainfall (i.e., by optimizing the method performances in properly reproducing rainfall amounts) by masking out days with no rainfall rate during the irrigation seasons (hence, potential irrigation days). The F parameter has been set equal to 0.3 as explained in Section 3.2. For further details on the method, the reader is referred to Dari et al. (2023).

R3C6. 5) Some of the main conclusions of this study (i.e., "Consequently, we can confidently stipulate that Sentinel-1 soil moisture data is capable of detecting the irrigation signal in space with good precision." or "The possible impact of COVID-19 pandemic is also highlighted.") are not fully supported, in my opinion, from the analysis, due to (a) the small range of data from 2015-2023 (at least 30 years of data is required to include the intrinsic uncertainty of the key hydrological-cycle processes, traced in their short-term dependence and long-term persistence), and (b) more data and different climatic and seasonal conditions need to be examined to support the above conclusions and exclude other factors that may result in the same impacts and similar images (for example, how do you take into account the type of irrigation and land-use in the area; is there a change during the 2015-2023 period?).

R3A6. The limited range of data is due to the temporal coverage of satellite data, which represents the essential tool on which the approach relies. We honestly believe that this is not a strong limitation to our study, as it is a natural issue in emerging applications relying on recent satellite retrievals. However, we will add the following sentence in the conclusions:

"The current temporal coverage of Sentinel-1-derived observations may be a limitation, but the continuity foreseen for the mission will offer the possibility of creating long-term time series of irrigation water use in the upcoming years."

It is important to highlight that the reliability of the retrieved irrigation amounts is consistent with precipitation dynamics. In the new "Study Area" section we will explain that the system has been working from 1990s onwards, adopting an automated network of centre pivot, thus reasonably implying a static scheme working through sprinkler irrigation. The recurrence of irrigated areas is also clear by the following maps of average LAI (Leaf Area Index) during the various irrigation seasons (see Figure R1 and Figure A4 in the Appendix to the manuscript).

[Figure]

**Figure R1**. Average LAI during the various irrigation seasons considered.